# FOSTERING EFFECTIVE COMMUNICATION BETWEEN HUMANS AND MACHINES

**Karthika Kamath, Tanya Upadhyay, Jieya Rawal, Kirtana Sunil Phatnani & Biju Dominic** [*]
Fractal Analytics,
Mumbai, Maharashtra 400063, India
{karthika.kamath,tanya.upadhyay,jieya.rawal,kirtana.phatnani,
biju.dominic}@fractal.ai

## ABSTRACT

With the growing usage of smartphones, digital communication has become significant. This paper describes the primary research conducted to study user interaction patterns on smartphones. Results show that the time between two touches, or the editorial context duration, is just 5 to 10 seconds for a vast majority of smartphone interactions. So, the paper introduces the novel concept of MicroStimuli, which can generate a response in mere milliseconds, specifically tailored for smartphones. The construct of MicroStimuli is formulated by leveraging the neuroscience of decision-making in response to visual stimuli.

## 1 INTRODUCTION

The fifth industrial revolution poses new challenges, where finding the synergy between humans and machines becomes the most important challenge (Noble et al., 2022). Given that the global smartphone penetration rate rose to 67% in 2021 (Laricchia, 2023), the most prevalent human-machine interface is most likely to be a smartphone screen. Hence, this paper explores the bounds of communication, for an always-on, always-with-you medium, smartphones.

It is assumed that the editorial context, in the traditional communication medium of television, remains constant for about 30 minutes, i.e., the average program length (Cha et al., 2008). So, the good old persuasion stimuli, a 30-second commercial, fits well within the context duration of the television medium. On smartphones, with every touch, the editorial context changes, especially on social media apps. Ideally, the ratio of the duration of the persuasion stimuli to the duration of the editorial context should be kept to the minimum, so that the persuasion stimuli is considered less of an intrusion (Riedel et al., 2023).

This paper describes the empirical research to understand the depths of smartphone interactions, and consequently, the editorial context duration on smartphones. This is followed by discussing the brain processes of decision-making in the final second of an E-commerce transaction on smartphones. This understanding is used to propose a possible framework that can help in developing effective digital persuasion stimuli for E-commerce.

## 2 EXPERIMENT SETUP

The authors reached out to a randomized convenience sample of students and working professionals in the age group of 20-30 years across India. Out of the 200 people who enrolled, 50 completed the 1-2 week-long study. This empirical research was conducted through a non-intrusive, context-immersed app on Android smartphones. The app stored the precise *timestamp* (as epoch time) and *app-name* for each touch (excluding keyboard typing). A detailed experiment methodology is provided in Appendix Section 1. Upon further analysis, out of 50 submissions, 44 users who had at least 7 days of recorded data, were shortlisted. A total of 443,520 minutes of smartphone activity and 1.8 million smartphone screen touches were analyzed.

---

[*]All authors have contributed equally to the study.

## 3 RESULTS AND ANALYSIS

**Time Spent and Number of Touches:** Experimental results show that the average user spends 3 hours and 36 minutes per day with their smartphone. During this time, 4513 is the median of the number of screen touches, across users, per day. For heavy users, the number of touches is as high as 15,040 in a day. The distribution of touches is shown in Appendix Table 1.

**Session Length:** It is observed that while participants spend a large amount of time on some applications, this total duration is segmented into multiple brief sessions. For example, 72 minutes are spent per day on Instagram. But this long duration, for Instagram, comprises 22 brief sessions per day. The average length of each of these Instagram sessions is 197 seconds. A similar pattern of brief, but multiple sessions throughout the day is observed for other popular apps such as WhatsApp, Facebook, YouTube, etc., as described in Appendix Table 2.

**Context Duration:** Studies show that the context duration for television is 5 to 21 minutes (90 to 95 % of viewing sessions) (Cha et al., 2008). In contrast, context duration, calculated as the time between two touches on smartphones, is less than 5 seconds for 90% of interactions and less than 10 seconds for 95% of interactions. Appendix Table 3 provides more detailed percentile distributions of context duration.

The authors propose the need to redesign the persuasion stimuli for smartphones that fit well into its much shorter context duration. This calls for the conception of MicroStimuli: persuasion stimuli that generates a response in only a few milliseconds.

The concept of MicroStimuli is the norm of Nature. Dutch biologist, Nikolaas Tinbergen, described *Supernormal stimuli*: exaggerated forms of natural stimuli that elicit *fixed action patterns* of responses in mere milliseconds, in several animal species (Tinbergen, 1951). In sports like tennis, baseball, cricket, or soccer, neuroscientists describe that the time available for decision-making, in the brain, is less than 0.4 seconds (McLeod, 1987). In sports, this knowledge of millisecond decision-making has been used to devise newer game strategies (like the slow ball, in cricket and change-up, in baseball). So, it is a common phenomenon seen in nature and human decision-making, to identify the patterns and suitably devise strategies to influence the decision-making process in milliseconds (Carter, 2019). In the next section, the authors present their hypothesis on how understanding decision-making on smartphones can be used for developing suitable MicroStimuli for a particular use-case of E-commerce purchase decisions.

## 4 DISCUSSION AND FUTURE DIRECTIONS

One of the measures to evaluate the effectiveness of any digital persuasion stimuli is Click Through Rate (CTR). Since the first digital ad in 1994, the average CTR has plummeted over 100 times, to just about 0.35% today (Hwang, 2020; Marino, 2023). Using the MicroStimuli, this challenge of low CTR in E-commerce can be mitigated. The MicroStimuli would be best developed by understanding what happens in a buyer's brain during the final second of the purchase decision. It has been found that the brain's first reaction to any stimuli is either to Approach or Avoid it (LeDoux & Bemporad, 1997). This Approach-Avoid decision is taken in a matter of 350 milliseconds (Carter, 2019). Categorization is the first stage in the brain while evaluating any stimuli (Murphy). 69% of the purchase decisions on Amazon.com begin with a category search term (Szahun & Dalton, 2021). Based on these findings we propose that triggering the 'core category need' of the product is ideal to facilitate the Approach response in the buyer's brain (Kotler & Keller, 2020). The next stage, Liking, is achieved by invoking past experiences with the brand (Berridge & O'Doherty, 2014; Berridge & Robinson, 2016). This happens in a matter of 100 milliseconds (Brielmann et al., 2017). Finally, the Wanting stage is activated by playing up the emotional benefits of the brand and strengthening the perception of price reward. This happens in a matter of 450 milliseconds (Braeutigam et al., 2001). The core category need, past experience utility, emotional benefit of the brand, and the price reward activate the Approach-Liking-Wanting process in the brain in 920 milliseconds. This becomes the foundation to develop an ideal MicroStimuli to enable purchase decisions in E-commerce. This framework is summarized in Fig. 1 (Upadhyay et al., 2022). The immediate next step, for the authors, is to test and establish the effectiveness of the proposed framework in a live scenario and scale up the development of MicroStimuli using generative AI tuned on the construct of this framework.

URM STATEMENT

The authors acknowledge that at least one key author of this work meets the URM criteria of ICLR 2023 Tiny Papers Track.

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

## A  APPENDIX

### A.1  DETAILED EXPERIMENTAL SETUP

The authors reached out to a randomized convenience sample of students and working professionals in the age group of 20-30 years across India. Out of the 200 people who enrolled, 50 completed the 1-2 week-long study. At the start of the study, participants provided informed consent. Post enrollment, they were asked to download an Android application, which identified each instance of a touch made on the smartphone screen and, optionally, the name of the application where the touch was made. To assuage privacy concerns, this application was not privy to on-screen content, within each of the applications, but only gathered the app name. Participants were also asked to customize two features on the app: (1) enable recording app-wise touches, (2) disable touch-counting when touches were made on the on-screen keyboard (to exclude typing). The app, thus, stored the precise *timestamp* (as epoch time) and *app-name* of each touch (excluding keyboard typing), on the user's own device, respecting the participants' privacy. At the end of the 2-week period, participants were asked to share a *.db* file, generated on their devices by the application. Each participant shared a single .db file. Each file shared was de-identified before processing. Upon careful analysis of data from the 50 participants, the authors selected 44 files that contained at least 7 consecutive days of touch data. The purpose of this study was to observe how people interact with their smartphones in a natural environment. Hence, participants were asked to use their smartphones as usual for the entire duration of the study, while the application ran in the background. The background application was non-intrusive in their daily activities.

### A.2  RESULTS

Table 1: Percentile distribution of the number of touches across smartphone users.

| Percentile | No. of touches |
|---|---|
| 25th perc. | 3160 |
| 50th perc. | 4513 |
| 75th perc. | 6167 |
| 90th perc. | 8615 |
| 95th perc. | 11068 |
| 99th perc. | 15040 |

Table 2: Distribution of time spent across the day, number of sessions per day, and time spent per session across popular apps.

| App name | Time spent per day (minutes) | No. of sessions per day | Time spent per session (seconds) |
|---|---|---|---|
| Facebook (FB) | 28 | 9 | 176 |
| Instagram (IG) | 72 | 22 | 197 |
| Amazon (Az) | 7 | 3 | 135 |
| Google Chrome (Ch) | 15 | 9 | 104 |
| Netflix (NF) | 19 | 4 | 283 |
| YouTube (YT) | 20 | 5 | 231 |
| WhatsApp (WA) | 51 | 35 | 83 |

Table 3: Percentile distribution of the context duration on smartphones.

| Percentile | Context duration (seconds) |
|---|---|
| 25th perc. | 0.43 |
| 50th perc. | 0.87 |
| 75th perc. | 1.98 |
| 90th perc. | 5.00 |
| 95th perc. | 10.00 |
| 99th perc. | 31.00 |

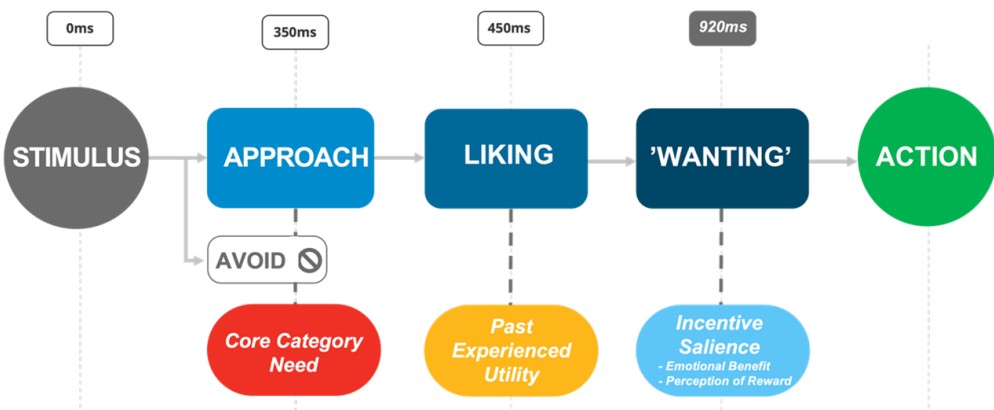

Figure 1: Composition of MicroStimuli with respect to brain processes in the Final Second of Decision Making.

