# OpenReview forum: "Fostering Effective Communication Between Humans and Machines"
_ICLR.cc/2023/TinyPapers — Submitted to Tiny Papers @ ICLR 2023_

### Official Review · Reviewer_z9rT · 2023-04-01

**Confidence:** 2

**Summary Of Contributions:**

This paper proposes a final second decision making framework (FSDM). This FSDM framework aims to understand how the brain understands and processes visual information in short time horizons such as in the context of smartphone use and makes actions from that.

**Rating:**

Needs Clarification (NC): a submission which does not meet the reviewing criteria and needs clarification for its described problem or solution

**Strengths And Weaknesses:**

Strengths:

- There is clear motivation detailed in the introduction regarding the need to tackle settings where visual stimulus time horizons are short and there are fast context shifts.

Weaknesses
- General paper's writing quality could be improved, some sentences are difficult to understand (see some examples in suggested changes)
- The introduction and discussion mention the utilization of the neuroscience of decision-making as a lens for approaching human machine interaction, however the paper only applies it in theory by citing a few papers and hypothesizing some clams in the secondary research section. It is unclear how neuroscience is really being applied to the core claims of this paper, and whether the secondary research section adds any value to those claims. Moreover, as a neuroscience paper the secondary research section should be revised to back up the claims behind the proposed components of the FSDM, these seem to be mostly postulations.
- In the introduction it is mentioned that the paper is inspired by past studies and so the approach is to find the minimum and maximum time (perhaps a better description would be the distribution of time) a visual stimulus on a smartphone is there for. However the section 2.2 results don't seem to say much about this distribution (or min/max times) other then mentioning the average. While the average is a useful metric, it would be useful to include a chart of the distribution of the measured times. Moreover, it is unclear how this is related to the FSDM other than clarifying that a decision making framework in a short time horizon context should focus on the final second. More clarification here would be great.
- It is unclear what the data is collected by this paper and how it contributes to any of the results/discussions. It seems data is pulled from another paper (Howarth, 2023) and the 2.7s average is from this paper. If this is the case, it would great to clarify what section 2.2's purpose is and the need for an experiment.
- The discussion section claims to investigate the relevant processes leading to an action (which as I understand is backed by section 3 on secondary research) but
- The content and contributions of the paper does not seem suitable for ICLR as it appears to synthesize data from another paper (Howarth, 2023) and discuss some potential implications of it. However, since this is the new Tiny paper's track I may misunderstand the requirements so if relevance to the ICLR community is not critical this comment can be ignored.

**Suggested Changes:**

Major:
- It would be important to include the distributions of data (averaged across all participants if needed) as a single average metric says little about the data and is difficult to make claims around
- The experiment results can be better utilized and explained how they support a claim
- I would recommend heavily revising the secondary research section by clarifying how it relates to the core claims


Minor:
- Section 1: "To begin understanding the time required to decide in the brain we take learnings from sports, where the batsman decides which shot to play in 200ms." Sentence could be reorganized e.g. "To understand the decision making time of the brain, we take learnings from sports such as Cricket where batsmen take 200ms to make a decision (McLeod, 1987)."
- Section 1: "Looking further we found Facebook (fac) and Netflix studies quoting that the time any user spends considering watching a
piece of content is just about 1.8 seconds (abo) in a browsing session." Seems to be incorrect citation / what does fac and abo mean?
- Section 3: "We propose that an Emotional Category-Need Trigger, e.g., thirst triggered by looking at visuals of a sandy desert, heat
waves, and condensation of water droplets on a chilled bottle of a soft drink." seems to be unfinished, what does this sentence want to say about proposing this?

---

> ### Author Response · Authors · 2023-05-31
> **We have included the distributions of data, better explained the experiment and its results and restructured the secondary section to better correlate with the core claims of the experiment**
>
> Response to Strengths:
>
> The authors thank the reviewer for their feedback on the Introduction Section having a clear motivation detailed for the need to tackle settings where visual stimulus time horizons are short and there are fast context shifts. We are honored!
>
> Response to Weaknesses:
>
> Thank you for taking your time to review our paper. We apologize for the poor writing quality and sentence structuring. We have re-written the paper and hope that it is more comprehensible now.
>
> In response to your feedback on the application of neuroscience to decision-making, as being more of claims, and the content of secondary research being posed more as postulations, we have restructured the paper to connect each section better and have stated explicitly where we are making a hypothesis. A new term of MicroStimuli is introduced based on the results of the experiment on context duration as explained in the ‘Results and Analysis’ section. The core idea of the earlier ‘Secondary Research’ section has been moved into ‘Discussion and Future Direction’ section in the revised paper. The application of neuroscience in the paper is limited to a particular use-case on smartphone - understanding the brain processes involved in making a purchase decision in E-commerce when observing a persuasion stimuli (an advertisement). This thought process is developed by taking learnings from multiple research papers to develop a possible framework. This framework, earlier addressed as FSDM has now been changed into a framework that helps develop the MicroStimuli. We agree that this framework has not been tested and is posed more like a theoretical framework which will be tested as the next step, as also highlighted in the last sentence of ‘Discussion and Future Directions’ section.
>
> In response to the third bullet point under Weaknesses, we understand it was a miscommunication and we have revised our ‘Introduction’ section to better communicate the core idea of the paper. We have also included the distribution of measured times (refer to the tables in the Appendix). One of the findings from the experiment was that the editorial context duration on smartphone for a vast majority of interactions is 5 to 10 seconds. So, the persuasion stimuli on smartphones should be a small fraction of this time such that the ratio of persuasion stimuli duration to editorial context duration is kept minimum. By keeping this ratio small the persuasion stimuli will be considered less of an intrusion. And we believe this can be achieved by going deeper into how the brain processes a digital visual stimulus and what should the constituents of the stimulus be to trigger the desired response in just a few milliseconds. This is the core idea tying the experiment results with our approach in developing a framework as explained in the Section 3 & 4 of the revised paper.
>
> In response to your fourth bullet point under weaknesses, we understand it was a miscommunication and have tried to resolve the confusion by describing the ‘Experiment Setup’ section better. We have also added an in-depth description in the Appendix Section 'Detailed Experiment Setup'.
>
> Response to suggested changes:
>
> Overall, we have tried to incorporate every feedback provided including the major and minor suggested changes, it was valuable to us. Your feedback helped us restructure the paper so that the core idea of the paper is presented effectively.

---

### Official Review · Reviewer_YYuW · 2023-04-01

**Confidence:** 2

**Summary Of Contributions:**

This paper explores how quickly humans tend to change context (e.g., by swiping) when interacting with media on a smartphone, which is substantially faster than context switches on, for example, television. Based on this, the paper argues that visual stimuli (such as ads) should be (re)designed in ways that account for this difference.

**Rating:**

Needs Clarification (NC): a submission which does not meet the reviewing criteria and needs clarification for its described problem or solution

**Strengths And Weaknesses:**

### Strengths
- The paper is well written and the general message is clear.
- While I'm not familiar with the economics- and neuroscience-based literature largely cited in this paper, it appears to me that the paper likely has a good overview of the related and relevant literature.

### Weaknesses
- I see no Machine Learning content in this paper. While I don't see an explicit list of topics that are considered in or out of scope for Tiny Papers, I assume that roughly the same topics would be considered relevant as for the main ICLR conference. If this is the case, I unfortunately think the topic of the paper is simply not within the scope for ICLR, and ICLR is not a suitable venue for this work.

**Suggested Changes:**

To be considered on-topic and relevant for ICLR, there should be at least some Machine Learning aspect in the paper.

---

> ### Author Response · Authors · 2023-05-31
> **We have addressed the lack of Machine Learning content in the paper.**
>
> Response to strengths:
>
> The authors thank the reviewer for sharing their confidence in the clarity of writing, concept, and the general message of this paper. The authors are honored that Respected Reviewer extends their trust to this paper’s overview and literature.
>
> Response to weaknesses:
>
> To the respected reviewer’s point about there being no machine learning content in the paper, we agree. This paper reflects on the interim findings in the process of marrying data and creative for developing MicroStimuli, as the paper goes on to suggest. The authors propose that in the brief context duration of smartphones, a 30-second commercial, or its edits, are not the most effective forms of visual communication. The current visual misrepresentation appears to be diminishing the interactivity between the information shown on smartphone screens, and the behaviors it is aimed at producing (for example, purchase decisions, or physical activity). The authors wonder how this human-machine interaction gap might be bridged. The first step in that direction (i.e., our research) is to study and understand the smartphone medium itself. A bottom-up approach to creation of visual representations is taken, as opposed to the norm of summarizing a 30-second commercial to fit a shorter format. The authors’ bottom-up approach to creative generation might be considered novel. The immediate next step, included in the closing statement, is to dynamically generate (with generative AI) these MicroStmuli, which is, otherwise, the painstaking, time-consuming role of a creative or a content designer, starting with a blank canvas.

---

### Author Response · Authors · 2023-06-13
**Opting-in for Archival**

We would like to opt-in for archival

---

### Meta-Review · Area_Chair_s5jV · 2023-04-08

**Recommendation:** Invite to revise
**Confidence:** 3

**Metareview:**

This paper explores how quickly humans tend to change context (e.g., by swiping) when interacting with media on a smartphone, which is substantially faster than context switches on, for example, television. It proposes a final second decision making framework (FSDM) to model this user behaviour.  that short horizon visual stimuli should be (re)designed in ways that account for this difference.

The paper is well-organized and has a clear motivation behind the idea being put forward. The overall idea of finding what is the attention span under which an user makes a decison is interesting and useful to tackle user persona problems in the HCI space.

However, the paper does not present any formative experiments/user studies to support the effectiveness of the FSDM framework. The current submission needs additional insights and more discussion around the claims made.

PS: We believe that the submission may not be contained under any particular topic of the ICLR Call for papers. WILL LEAVE FOR THE PCs TO DECIDE ON RELEVANCE.

Overall, notwithstanding relevance, we recommend the authors to be invited to revise based on suggested changes.

**Summary:**

This paper explores how quickly humans tend to change context (e.g., by swiping) when interacting with media on a smartphone, which is substantially faster than context switches on, for example, television. It proposes a final second decision making framework (FSDM) to model this user behaviour. The paper needs additional insights and clarifications on the experiments and claims mentioned in the paper, and also how they reached this framework suggested.

**Comments And Feedback To The Authors:**

The paper presents an interesting insight into how the user context switches have become much more rapid with smartphones as compared to televisions. With this comes the need to adapt our modelling of the user.

There are some revisions suggested for improvements to the paper :
1. It would be important to include the distributions of data to make more generalizable claims.
2. The experiment results can be better utilized and explained how they support a claim.
3. We would recommend heavily revising the secondary research section by clarifying how it relates to the core claims.
4. Rectify typos and use macros to standardize writing wherever possible.

**Reason For Not Giving A Higher Recommendation:**

There are suggested changes needed to make the claims convincing and paper archivable.

**Reason For Not Giving A Lower Recommendation:**

N/A

---

> ### Author Response · Authors · 2023-05-31
> **We have included the distributions of data, better explained the experiment and its results and restructured the secondary section to better correlate with the core claims of the experiment**
>
> Response to metareview:
>
> Yes, we agree that our paper reports a study on how the medium of the smartphone is substantially faster than the television. We thank the reviewer for finding our paper well-organized and with clear motivation. We are honored!
> The authors sincerely thank the reviewer for their valuable feedback on the effectiveness of the framework (earlier FSDM), additional insights, and discussions around the claims made. To that effect, we have made our primary research (described in the Introduction, Experimental Setup, and Results and Analysis Section), i.e., understanding the context duration on smartphones, the centerpiece of the study.
>
> Our findings conclude that the context duration on smartphones is merely 5 to 10 seconds (90th and 95th percentile). To create a persuasion stimuli on smartphones that is non-intrusive, we define the concept of a MicroStimuli which works in a matter of few milliseconds. In the Discussion and Future Directions Section, we elaborate on how the MicroStimuli can be constructed for a purchase decision through a framework (earlier FSDM). Furthermore, we provide a detailed description of the framework (earlier FSDM). We propose the next steps of this research to be, testing the effectiveness of the proposed framework of MicroStimuli for a purchase decision in a live scenario, and scaling up the development of MicroStimuli using generative AI tuned on the construct of this framework.
>
> The authors thank the respected reviewer for their valuable feedback on the revisions required for this paper. To that effect, we have rewritten the entire manuscript to better explain:
>
>  - How the smartphone medium needs an understanding that is vastly different from the television medium?
>  - How understanding the editorial context duration of smartphones allows us to identify the time available for a persuasion stimuli, that is - non-intrusive?
> - How can we design a MicroStimuli that works in such a short context duration?
>
>
> Response to Summary:
>
> The authors thank the respected reviewer for their valuable feedback on the experiments and claims mentioned in the paper and how we arrived at the suggested framework. To that effect, we have rewritten the entire manuscript elaborating:
> - In the Introduction Section: the need for understanding the always-on, always-with-you medium of smartphones which is vastly different from the television medium.
> - In the Experimental Setup Section: a detailed description of the demographics of the subjects, the duration of the study, details of the resources used to gather the data for this analysis, and how was the data pre-processed for the analysis. We have further provided a detailed experimental setup in the appendix section.
> - In the Results and Analyses Section: We discuss the contrast between the context duration of the television medium and the smartphone medium. We further provide detailed data gathered regarding the distribution of the number of touches in a day, app-wise session durations, and the distribution of context duration in Tables 1, 2, and 3. From this, we conclude the need for a MicroStimuli.
> - In the Discussions and Future Directions: We have elaborated on how a MicroStimuli can be constructed for an E-commerce purchase decision by understanding the brain processes involved. We further provide strong references to each of the brain processes. We also identify what component in the stimuli would trigger these brain processes.
>
>
> Response to comments and feedback to the authors:
>
> - (1 & 2) The authors thank the reviewer for their feedback on providing the distributions of the data to make more generalizable claims. To that effect, we have provided the distribution of touches across users in a day in Table 1 and the distribution of context durations in a day for an average user in Table 1. We have also provided how the variation of session duration happens across various apps in Table 2. We have further elaborated on how in order to create a non-intrusive persuasion stimulus on smartphones, we need a MicroStimuli. A MicroStimuli works in a matter of just few milliseconds. We further describe how the MicroStimuli is the norm in Nature.
> -  (3) The authors thank the reviewer for their feedback on revising the Secondary Research Section and clarifying how it relates to the core claims. To that effect, we have completely revised the manuscript. To further clarify how the framework connects to the core claims, in the Results and Analyses Section, we establish the need to construct a MicroStimuli ideal for the smartphone medium. In the Discussion and Future Directions Section, we describe how understanding the brain processes in the final seconds of a purchase decision lends insights into the construction of a MicroStimuli for improving CTRs.
> - (4) The authors thank the reviewer for their feedback for correcting typos and using macros wherever possible. To that effect, we have rewritten the manuscript and rectified all possible typos.

---

### Decision · Program_Chairs · 2023-04-09

**Decision:**

Revision accepted; invite to archive

**Comment:**

This paper and topic falls within scope, and the authors are asked to revise according to the reviewers' feedback.